# The Adenoviral E1B-55k Protein Present in HEK293 Cells Mediates Abnormal Accumulation of Key WNT Signaling Proteins in Large Cytoplasmic Aggregates

**DOI:** 10.3390/genes12121920

**Published:** 2021-11-29

**Authors:** Petter Angell Olsen, Stefan Krauss

**Affiliations:** 1Hybrid Technology Hub-Centre of Excellence, Institute of Basic Medical Sciences, University of Oslo, 0317 Oslo, Norway; stefan.krauss@medisin.uio.no; 2Unit for Cell Signaling, Department of Immunology and Transfusion Medicine, Oslo University Hospital, P.O. Box 4950, Nydalen, 0424 Oslo, Norway

**Keywords:** WNT signaling, HEK293 cells, E1B-55k, cytoplasmic aggregates

## Abstract

HEK293 cells are one of the most widely used cell lines in research, and HEK293 cells are frequently used as an in vitro model for studying the WNT signaling pathway. The HEK293 cell line was originally established by transfection of human embryonic kidney cells with sheared adenovirus 5 DNA, and it is known that that HEK293 cells stably express the adenoviral E1A and E1B-55k proteins. Here, we show that HEK293 cells display an unexpected distribution of key components of the WNT/β-catenin signaling pathway where AXIN1, APC, DVL2 and tankyrase are all co-localized in large spherical cytoplasmic aggregates. The cytoplasmic aggregates are enclosed by a narrow layer of the adenoviral E1B-55k protein. The reduction of E1B-55k protein levels leads to the disappearance of the cytoplasmic aggregates thus corroborating an essential role of the E1B-55k protein in mediating the formation of the aggregates. Furthermore, HEK293 cells with reduced E1B-55k protein levels display reduced levels of transcriptional activation of WNT/β-catenin signaling upon stimulation by the Wnt3A agonist. The demonstrated influence of the E1B-55k protein on the cellular localization of WNT/β-catenin signaling components and on transcriptional regulation of WNT/β-catenin signaling asks for caution in the interpretation of data derived from the HEK293 cell line.

## 1. Introduction

The WNT/β-catenin signaling pathway plays a pivotal role during development, organogenesis, stem cell maintenance and self-renewal of many tissues, and aberrant WNT/β-catenin signaling activity is associated with the development of several disease conditions including cancer [1,2]. The main determinant of WNT/β-catenin signaling activity is the level of free cytoplasmic β-catenin that can translocate to the nucleus and activate target genes through interactions of β-catenin with transcription factors of the TCF/LEF family [3]. In cells with low WNT/β-catenin signaling activity, the level of cytoplasmic β-catenin is kept low by the constant action of a degradosome that targets cytosolic β-catenin (by phosphorylation and ubiquitination) for proteolytic degradation. The degradosome comprises in its core of the structural protein APC, the rate limiting structural proteins AXIN1/2, β-catenin, the kinases CK1 and GSK3β and the ubiquitin ligase β-TrCP [4]. A further protein, tankyrase (TNKS), may associate with the degradosome and regulate its activity by controlling the stability of the AXIN1/2 proteins [5,6]. The activation of WNT/β-catenin signaling by binding of an exogenous WNT ligand, such as WNT3A, to the frizzled (Fz) and low-density lipoprotein receptor-related protein (LRP5/6) receptors at the cell membrane leads to interference with degradosome activity, thus resulting in increased levels of cytosolic β-catenin [7]. The typical model for degradosome inactivation upon WNT ligand receptor binding involves disassembly of the degradosome and formation of a signalosome due to interactions between AXIN1/2 and DVL2 at activated Fz and LRP5/6 membrane receptors [8,9].

Cell lines are important tools in cell biology research, and one of the most commonly used cell lines is human embryonic kidney (HEK) 293 cells [10,11]. The HEK293 cell line was originally established by transfection of primary human embryonic kidney cells with sheared adenovirus 5 DNA, and it has been shown that HEK293 cells stably express the adenoviral E1A and E1B-55k proteins due to integration of a 4 kbp adenoviral DNA fragment in chromosome 19 [11,12]. In the context of WNT/β-catenin signaling, HEK293 cells are frequently used as a model system since they do not contain known mutations in central WNT/β-catenin signaling components and have a functional WNT/β-catenin signaling cascade from the cell surface to the nucleus. Thus, HEK293 cells display low levels of basal canonical WNT/β-catenin signaling activity that efficiently can be stimulated with exogenous WNT ligands, such as WNT3A, leading to the increased transcription of WNT/β-catenin target genes [13,14,15].

Given the broad use of HEK293 cells in WNT signaling research, we characterized the subcellular localization of key components of the WNT/β-catenin signaling pathway. We show that while β-catenin is predominantly localized at the cell membrane, AXIN1, APC, DVL2 and TNKS all co-localize in large cytoplasmic aggregates that are enclosed by a narrow layer of the adenoviral E1B-55k protein present in HEK293 cells. We further analyzed E1B-55k protein interaction partners and identified the AMER1 protein as a potential link between WNT signaling components and the E1B-55k protein in HEK293 cells. Finally, we demonstrated that siRNA-mediated reduction of E1B-55k protein levels lead to a dispersal of the cytoplasmic bodies and that HEK293 cells with reduced E1B-55k protein levels display reduced WNT/β-catenin transcriptional activation upon stimulation with Wnt3A.

## 2. Materials and Methods

### 2.1. Cell Lines

The human embryonic kidney cell lines HEK293 (ATCC CRL-1573) and HEK293T (ATCC CRL-3216) were obtained from the American Type Culture Collection (Manassas, VA, USA). The stable reporter HEK293-STF/Ren cell line was described previously [16]. All cell lines were cultured in DMEM (Merck) supplemented with 10% FBS (Thermo Fisher Scientific, Waltham, MA, USA) and 1% penicillin/streptomycin (Merck, Darmstadt, Germany) at 37 °C in a humidified atmosphere containing 5% CO_2_.

### 2.2. Manipulation of WNT Signaling Activity by Wnt3A CM and G007-LK Treatment

To stimulate or inhibit WNT signaling activity, cells were treated for 24 h with Wnt3A CM or G007-LK, respectively. Wnt3A CM was prepared by collecting the supernatant from L Wnt3A cells (ATCC CRL-2647) according to the manufactures protocol. Unless otherwise stated, HEK293 cells were treated with Wnt3A CM diluted 50% in complete DMEM. Stock solutions of G007-LK [16] (and Selleck Chemicals, Houston, TX, USA) was prepared in DMSO and was added to the growth medium at a final concentration of 1 µm.

### 2.3. Immunofluorescence Staining and Microscopy

Cells grown on coverslips precoated with poly-L-lysine (Santa Cruz Biotechnology, Dallas, TX, USA) were fixed in 4% paraformaldehyde (Merck) (15 min at room temperature (RT)) permeabilized with 0.1% Triton-X100 (Merck)/PBS (15 min at RT) followed by incubation with primary and secondary antibodies diluted in PBS with 4% bovine serum albumin (Merck) (1 h RT each). Nuclear counterstaining was performed with DAPI (Merck) (1 μg/mL, 5 min RT), and the coverslips were mounted in ProLong Diamond Antifade Mountant (Thermo Fisher Scientific). The following primary antibodies and dilutions were used: β-catenin (BD Biosciences; BD610153; 1:500), Tankyrase-1/2 (H-350) (Santa Cruz Biotechnology; sc-8337; 1:50), Tankyrase-1/2 (E10) (Santa Cruz Biotechnology; sc-365897; 1:40), AXIN1 (C76H11) (Cell Signalling Technology, Danvers, MA, USA; #2087; 1:50), APC (F-3) (Santa Cruz Biotechnology; sc-9998; 1:50), DVL2 (10B5) (Santa Cruz Biotechnology; sc-8026; 1:50), E1B-55k (2A6) (Generous gift from P.E. Branton [17]; 1:50), SV40 T Antigen (Ab-2) (Merck; DP02; 5 μg/mL). Secondary antibodies (all from Thermo Fisher Scientific; 1:500): anti-Rabbit IgG Alexa488 (A-21206), anti-Mouse IgG Alexa488 (A-11001), anti-Rabbit IgG Alexa594 (A-11012), anti-Mouse IgG Alexa594 (A-11005), anti-Rabbit IgG Alexa647 (A-21246), anti-Mouse IgG Alexa647 (A-31571). Fluorescent images were acquired with a Zeiss Elyra PS1 microscope system (Carl Zeiss Microscopy, Jena, Germany) using standard filters sets and laser lines with a Plan-APOCHROMAT 63x 1.4 NA oil objective. Images were acquired using the “Laser wide field (WF)” or “Structured Illumination Microscopy (SIM)” mode of the system as indicated. SIM images are specified in the figure legends. WF images were acquired for 30 Z planes and are displayed as maximum intensity projections rendered from all Z planes. SIM images were acquired using 5 grid rotations for 22 Z planes with a Z spacing of 0.184 nm between planes. SIM images were reconstructed with the following “Method” parameters in the ZEN black software (Carl Zeiss Microscopy): Processing: manual; Noise Filter: −5.5; SR Frequency Weighting: 1; Baseline Cut; Sectioning: 100/83/83; Output: SR-SIM; PSF: Theoretical. Unless otherwise mentioned SIM images are displayed as maximum intensity projections rendered from all Z planes.

### 2.4. Co-Immunoprecipitation (Co-IP)

Total HEK293 cell lysates were prepared by lysing cells in ice-cold lysis buffer (50 mM Tris pH 7.5, 150 mM NaCl, 1.5 mM MgCl2, 0.5% Nonidet P-40), and complete protease inhibitor cocktail (Merck) for 20 min on ice. For Co-IP, Dynabeads protein G (Thermo Fisher Scientific) was used according to the manufacturer’s instructions. In brief, antibodies (control normal mouse IgG (Santa Cruz Biotechnology; 2 ug) and (E1B-55k (2A6) (Generous gift from P.E. Branton [17]; 10 μL)) were bound to 50 μL washed beads by incubation for 1 h at room temperature. The antibody-coated beads were then incubated with cell lysate (400 μg total protein) at 4 °C overnight. The beads with immune complexes were washed four times with lysis buffer and two times with PBS/0.02% Tween-20 (Merck) and resuspended in PBS before analysis by liquid chromatography-tandem mass spectrometry (LC-MS/MS).

### 2.5. LS-MS/MS Analysis

Beads from the Co-IP were dissolved in 20 µL 0.2% ProteaseMAX Surfactant, Trypsin Enhancer in 50 mm NH4HCO3 followed by protein reduction, alkylation and on-beads digestion with trypsin overnight in 37 °C. After digestion, the samples were centrifuged at 14,000× *g* for 10 min, trypsin was inactivated by adding 100 µL 1% TFA, and the samples were again centrifuged at 14,000× *g* for 10 min. The resulting peptides were desalted and concentrated before mass spectrometry by the STAGE-TIP method using a C18 resin disk (3M Empore). Each peptide mixture was analyzed by a nEASY-LC coupled to QExactive Plus (ThermoElectron, Bremen, German) with EASY Spray PepMap^®^RSLC column (Thermo Fisher Scientific) (C18, 2 µL, 100Å, 75 µm × 50 cm). The resulting MS raw files were submitted for protein identification using Proteome Discoverer 2.1 (Thermo Fisher Scientific) and Mascot 2.5 (MatrixScience, London, UK) search engines. The search criteria for Mascot searches were: trypsin digestion with two missed cleavage allowed, carbamidomethyl (C) as fixed modification and acetyl (N-term), Gln->pyro-Glu (N-term Q), and oxidation (M) as dynamic modifications. The parent mass tolerance was 10 ppm and MS/MS tolerance 0.1 Da. The SwissProt database was used for the database searches. All of the reported protein identifications were statistically significant (*p* < 0.05) in Mascot and filtered in Proteome Discoverer Software (Thermo Fisher Scientific) for at least medium confidence identifications.

### 2.6. siRNA Transfection

Cells were transfected with the indicated siRNA construct (40 nM final concentration) using the Lipofectamine 2000 transfection reagent (Thermo Fisher Scientific) according to the manufacturer’s instructions. The transfection mixture was kept on the cells for 24 h followed by growth medium replacement and incubation for another 48 h before fixation for immunofluorescence or preparation of cell lysates for Western blot analysis. The following siRNA sequences were used (all from Merck): Control siRNA: endoribonuclease-prepared siRNA pool (esiRNA) that that target multiple locations in the EGFP mRNA sequence (Merck; EHUEGFP), E1B-55k siRNA (I): (GGA GCG AAG AAA CCC AUC UGA UU) and E1B-55k siRNA (II): (GGC CAG AUU GCA AGU ACA AGA UU). E1B-55k siRNA constructs I and II targets nucleotide positions 3–23 and 542–562 in the E1B-55k CDS, respectively (from HAdV5 GenBank accession no. AY339865.1: 2019–3509).

### 2.7. Quantification of Spherical TNKS Bodies Number in Cells

Quantification of average number of spherical TNKS bodies per cell was performed by counting number of spherical TNKS bodies per cell per cell in 10 randomly acquired images from control and E1B-55k siRNA-treated cells (>100 cells counted per treatment).

### 2.8. Western Blot Analysis

Total cell lysates were prepared by lysing cells ice cold RIPA buffer (Merck) supplemented with complete protease inhibitor cocktail for 20 min on ice. Protein concentrations were quantified by the Pierce BCA Protein Assay kit (Thermo Fisher Scientific). Lysates (20 µg total protein) were electrophoresed (NuPAGE Novex 3–8% tris-acetate protein gels (Thermo Fisher Scientific)) and transferred to PVDF membranes (Merck). Following blocking of the PVDF membranes (5% nonfat dry milk (AppliChem, Darmstadt, DE) in TBST, 1 h room temperature), immunodetection of proteins was performed by incubation with primary antibodies overnight at 4 °C and incubation with secondary antibodies for 1 h at room temperature. The following primary antibodies and dilutions were used: Tankyrase-1/2 (E10) (Santa Cruz Biotechnology; sc-365897; 1:200), AXIN1 (C76H11) (Cell Signaling Technology; #2087; 1:1000), DVL22 (30D2) (Cell Signaling Technology; #3224; 1:1000), E1B-55k (2A6) (Generous gift from P.E. Branton [17]; 1:5000), β-catenin (BD Biosciences, Franklin Lakes, US; BD610153; 1:5000), actin (Merck, A2066, 1:4000). Secondary antibodies (all from Santa Cruz Biotechnology; 1:5000): chicken antirabbit IgG-HRP (sc-2955), chicken antimouse IgG-HRP (sc-2954). Chemiluminescent detection was performed with the ECL Prime Western blotting Detection Reagent (Amersham–GE Healthcare, Chicago, IL, USA) and the ChemiDoc Touch Imaging System (Bio-Rad, Hercules, CA, USA). Quantification of E1B-55k protein levels as determined by calculating the ratio of the background subtracted intensities of the respective protein bands relative to the corresponding actins bands (Image Lab Software, Bio-Rad).

### 2.9. Supertop Flash (STF) Luciferase Reporter Gene Assay

Analysis of the effect of the E1B-55k protein on WNT signaling was performed using the HEK293-STF/Ren cell line where the β-catenin responsive TCF/LEF driven reporter (7xTCF/LEF binding sites regulating Firefly luciferase expression) and the constitutive CMV Renilla luciferase constructs were stably integrated [16]. One day after plating in 6-well dishes, HEK293-STF/Ren cells were left untreated or transfected with control or E1B-55k siRNA. In addition, 24 h after siRNA transfection, the cells were split, distributed in 12 well dishes and incubated for 24 h before the growth medium was replaced with medium containing increasing amounts (0, 2, 10 and 30%) of Wnt3A-conditioned media (CM). After 24 h incubation with Wnt3A CM the cells were harvested in passive lysis buffer and both Firefly and Renilla luciferase activities were measured in the same sample using the dual-luciferase reporter assay system (Promega, Madison, WI, US). The luminescence was quantified on a GloMax-Multi microplate reader (Promega) according to the manufacturer’s instructions. For figure preparation, the ratios of Firefly luciferase to Renilla luciferase signals were used.

### 2.10. Statistical Analysis

Testing for statistical significant difference between means was performed using the two-tailed unpaired *t*-test or one-way ANOVA as indicated in the figure legends.

## 3. Results and Discussion

### 3.1. Key WNT Signaling Components Are Localized in Large Cytoplasmic Aggregates in HEK293 Cells

To characterize the subcellular localization of central WNT/β-catenin signaling components, immunofluorescence (IF) staining was performed on untreated HEK293 cells and on cells incubated either with Wnt3A conditioned medium (CM) to activate the WNT signaling pathway or with the TNKS inhibitor G007-LK to inhibit WNT signaling activity (due to stabilization of the degradosome [18,19,20]). IF staining of β-catenin confirmed its established localization pattern: In untreated HEK293 cells, the majority of β-catenin was predominantly located at the cell membranes (Figure 1, row I, left column). The treatment of cells with Wnt3A CM resulted in increased levels of β-catenin both in the cytoplasm and in the nucleus (Figure 1, row I middle column). In cells treated with G007-LK, the localization of β-catenin was unchanged relative to what was observed in untreated cells (Figure 1, row I, right column). Interestingly, IF staining of AXIN1, APC and DVL2 displayed an accumulation in large cytoplasmic perinuclear aggregates that also contained TNKS (Figure 1, rows II, III and IV). Typically, each cell contained only one of these large cytoplasmic aggregates. We were not able to detect AXIN2 in HEK293 cells by IF (not shown) which probably reflects the low expression levels of AXIN2 in these cells (2.9–4.8 transcripts-per million [21,22]). The treatment of cells with either Wnt3A or G007-LK did not influence the cytoplasmic localization or amount of APC or DVL2 (Figure 1, row III and IV, middle and right columns). However, both AXIN1 and TNKS displayed increased fluorescent staining intensity following G007-LK treatment, thus indicating G007-LK mediated stabilization of AXIN1 and TNKS protein levels (Figure 1, row II, right column) as previously demonstrated upon TNKS inhibition in HEK293 cells [23,24,25,26].

### 3.2. The Cytoplasmic Aggregates Are Organized in Spherical Structures Enclosed by a Narrow Layer of the Adenoviral E1B-55k Protein, which Is Required for Aggregate Formation

The subcellular localization and size of β-catenin degradosomes have been reported to vary between experimental systems and cell lines [4]. However, the accumulation of central WNT/β-catenin signaling components in a single large cytoplasmic body as observed in the HEK293 cell line has not been described in other cell lines. The HEK293 cell line was originally established by transfection with sheared adenoviral DNA, and it has been shown that these cells express the adenoviral E1A and E1B proteins [11,12]. The adenoviral E1B-55k is a multifunctional protein that interacts with numerous viral and cellular factors [27]. In adenovirus-infected cells, the E1B-55k protein typically assembles with the viral E4 Orf6 protein along with cellular proteins to form a virus specific E3 ubiquitin ligase that regulates the degradation of cellular proteins such as TP53, MRE11, DNA ligase IV and BLM that constrain viral replication. It has further been demonstrated that in E1A and E1B transformed cell lines such as HEK293 (which lacks E4 Orf6 protein) while E1A displays a nuclear localization, the E1B-55k protein commonly localizes in large cytoplasmic aggregates. In HEK293 cells, these aggregates have been shown to contain various proteins including TP53, WT1, SSB2 and the MRN complex and that the sequestering of these proteins in E1B-55k aggregates perturbs normal functions of the proteins [28,29,30,31,32,33,34]. To investigate if the observed cytoplasmic degradasome aggregates in HEK293 cells co-localized with the E1B-55k protein, IF staining of the TNKS and E1B-55k proteins was performed. As seen in Figure 2, the majority of E1B-55k protein indeed accumulated in large cytoplasmic aggregates that co-localized with the TNKS stained cytoplasmic bodies. Although small clusters of E1B-55k that did not co-localize with TNKS were also present, all TNKS positive bodies displayed co-localization with the large E1B-55k protein aggregates. A commonly used variant of the HEK293 cells is the HEK293T cell line that stably expresses the SV40 large T-antigen [35]. In addition, HEK293T cells were confirmed to express the E1B-55k protein, and similar to what was observed in HEK293 cells, IF analysis of HEK293T cells showed an analogous accumulation of AXIN1, APC, DVL2 and TNKS in large cytoplasmic bodies (Appendix A). Given that other HEK293-derived cell lines such as 293S, 293SG, 293SGGD and 293FTM (Flp-in) all have conserved the genomic integration of the E1A/E1B encoding adenoviral sequences [10], these cell lines are also likely to encompass the large E1B-55k containing protein clusters as observed in the HEK293 and HEK293T cell lines.

To investigate a possible role of the E1B-55k protein in the formation of the cytoplasmic aggregates and to gain insight into the protein organization in the TNKS/E1B-55k aggregates, siRNA was used to reduce E1B-55k protein levels, and high-resolution images were acquired by structured illumination microscopy (SIM) [36]. SIM imaging of E1B-55k and TNKS proteins in HEK293 cells treated with control siRNA revealed that the cytoplasmic TNKS aggregates were assembled in spheroid structures composed of a central TNKS core encapsulated by a narrow layer of E1B-55k protein (Figure 3a) (hereafter referred to E1B-55k aggregates). In Appendix A, an animated three-dimensional rendering of a TNKS stained aggregate enclosed with the E1B-55k protein layer is shown. The E1B-55k aggregates varied in size with diameters ranging from 1 to 4 μm (not shown), and the width of the E1B-55k protein layer surrounding TNKS was measured to be around 220 nm (Appendix A). The treatment of HEK293 cells with two independent E1B-55k siRNAs efficiently reduced the E1B-55k protein levels to 29% and 18% 72 h after transfection (Appendix A). E1B-55k siRNA treatment led to a disappearance of the spherical TNKS aggregates in the majority of cells (Figure 3b), thus reducing of the average number of TNKS aggregates per cell from 0.75 to 0.18 (Figure 3c). Some E1B-55k protein could still be detected in E1B-55k siRNA-treated cells; however, rather than enclosing TNKS in spherical arrangements, the remaining E1B-55k protein formed rod-like structures (Figure 3b). The TNKS protein displayed a uniform cytoplasmic localization in the E1B-55k siRNA-treated cells, although some of the TNKS protein co-localized with the remaining E1B-55k rod-like structures (Figure 3b).

In the E1B-55k, siRNA-treated cells as well as the cellular distribution of AXIN1, APC and DVL2 were changed from the accumulation in the E1B-55k aggregates to a uniform cytoplasmic distribution (Appendix A). These results establish that the E1B-55k protein mediates the formation and accumulation of central WNT/β-catenin components in large cytoplasmic aggregates in HEK293 cells.

### 3.3. Analysis of E1B-55k Interaction Partners in HEK293 Cells

To identify E1B-55k interaction partners in HEK293 cells and to detect possible links between the E1B-55k protein and components of the WNT/β-catenin signaling pathway, co-immunoprecipitation (Co-IP) was carried out. Total HEK293 cell extracts were immune-precipitated (IP) with anti-E1B-55k antibodies and proteins present in the immunoprecipitates were identified by liquid chromatography-tandem mass spectrometry (LS-MS/MS) in two independent experiments (Appendix A). The list of proteins that were detected in both experiments included previously well-established E1B-55k interaction partners such as TP53 and components of the MRN DNA repair complex (MRE11, RAD50 and NBN) [37,38]. An interaction of E1B-55k with TP53, as detected in the IP of HEK293 cells, has been shown in several cellular systems and sequestration of TP53 in E1B-55k clusters is seen as a way for adenoviruses to neutralize TP53 activation during infection [27,39]. Although not well established, interactions between TP53 and the WNT signaling components AXIN1, DVL2 and GSK3β have been reported [40,41,42,43,44,45]. Additional possible candidates for mediating a direct link between E15-55k and WNT/β-catenin signaling pathway components included the APC and AMER1 proteins that were detected as E1B-55k interaction partners in one of the Co-IP experiments. APC is a large multifunctional protein with scaffolding functions that is essential for the assembly of the β-catenin destruction complex [46]. An association between APC and E1B-55k would be compatible with the observed clustering of key WNT signaling components in the E1B-55k aggregates; however, APC has not been detected as a direct E1B-55k interaction partner in previous studies [37,38]. Conversely, AMER1 (also called WTX) has been shown to interact both with the β-catenin destruction complex [47,48] and with the E1B-55k protein [49] (including in HEK293T and HEK293 cells, respectively). AMER1 regulates WNT signaling activity through its role as a scaffold protein for the β-catenin destruction complex [50] and would accordingly represent a feasible candidate protein for mediating the sequestering of WNT signaling components in the E1B-55k clusters. Nonetheless, a conclusive identification of the link between the E1B-55k protein and key components of the WNT signaling pathway requires validation by additional experiments that are outside the scope of this work.

### 3.4. Reduction of E1B-55k Protein Levels in HEK293 Cells Causes Decreased WNT/β-Catenin Mediated Transcriptional Activation upon Wnt3A Agonist Treatment

Given the observed E1B-55k-mediated accumulation of TNKS, AXIN1, APC and DVL2 in cytoplasmic aggregates (Figure 3), we next examined the impact of the cytoplasmic E1B-55k aggregates on WNT/β-catenin signaling. First, the protein levels of β-catenin, AXIN1, DVL2 and TNKS were analyzed after E1B-55k siRNA treatment. As seen in Figure 4a, E1B-55k knockdown did not have substantial influence on the total protein levels of any of these proteins thus indicating that the E1B-55k protein did not affect their expression levels or stability. Next, the effect of E1B-55k siRNA treatment on β-catenin-mediated transcriptional activation was tested using a HEK293 cell line incorporating the well-established SuperTopFlash (STF) reporter system [51]. The HEK293-STF/Ren cell line has a genomically integrated β-catenin responsive STF Firefly luciferase reporter gene together with a constitutive active control Renilla luciferase reporter gene and represents a robust assay for quantifying changes in WNT/β-catenin signaling activity [16]. When control siRNA-transfected HEK293-STF/Ren cells were incubated with increasing amounts (0–2–10–30%) of Wnt3A CM (to stimulate WNT/β-catenin signaling), the STF reporter was activated in a dose-dependent manner reaching a maximum of 23-fold induction with 30% Wnt3a CM (compared to unstimulated cells) (Figure 4b). HEK293-STF/Ren cells transfected with E1B-55k siRNA also displayed a dose-dependent activation of STF by Wnt3A, although in these cells the level of STF activation only reached a maximum of 12-fold induction with 30% Wnt3A CM. A significant reduction in reporter gene activation in E1B-55k siRNA-treated cells, relative to control siRNA-treated cells, was also observed upon stimulation with 10% Wnt3A CM (Figure 4b). Thus, in HEK293 cells with low E1B-55k protein levels, the β-catenin-dependent transcriptional activation by Wnt3A was significantly reduced compared to what was observed in HEK293 cells with normal E1B-55k levels. The observed increased potency of WNT/β-catenin signaling activation in HEK293 cells with native E1B-55k protein levels would be compatible with a scenario where the aggregation of central components of the β-catenin destruction complex in the E1B-55k clusters interferes with the destruction complex activity resulting in altered WNT/β-catenin signaling activation.

## 4. Conclusions

In this work, we showed that in HEK293 cells central components of the WNT/β-catenin signaling pathway display atypical cellular localizations by being sequestered in large cytoplasmic E1B-55k aggregates. By super-resolution microscopy, we established that these aggregates have a spherical structure that is surrounded by a narrow layer of E1B-55k protein. We found that reducing E1B-55k protein levels leads to the disappearance of the cytoplasmic aggregates, thus corroborating an essential role of the E1B-55k protein in mediating the formation of the aggregates. Lastly, our study reveals that HEK293 cells with reduced E1B-55k protein levels display diminished transcriptional activation of a WNT/β-catenin signaling reporter upon Wnt3A stimulation, thus demonstrating a functional effect of the aggregation on WNT signaling activity. The established impact of E1B-55k on both the cytoplasmic localization of key components of the WNT/β-catenin pathway and on the activity of WNT/β-catenin signaling in HEK293 cells asks for caution in the interpretation of data derived from this standard WNT signaling workhorse cell line.

## Figures and Tables

**Figure 1 genes-12-01920-f001:**
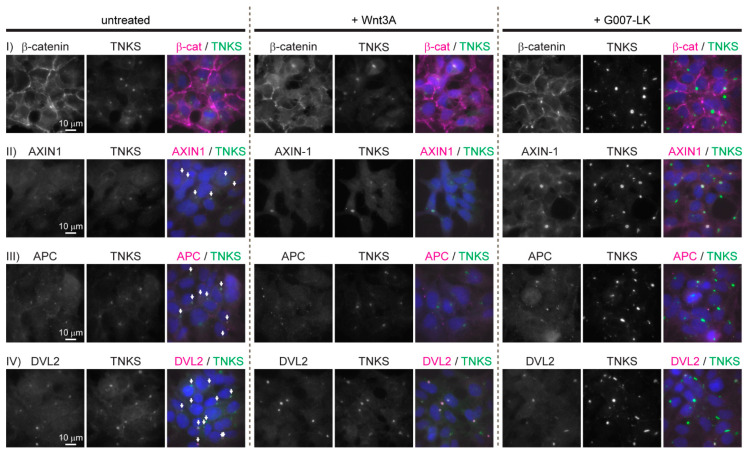
Key WNT/β-catenin signaling components localizes in large cytoplasmic aggregates in HEK293 cells. Representative fluorescent images (wide field (WF)) of untreated, Wnt3A- and G007-LK-treated HEK293 cells immunostained for β-catenin, TNKS, AXIN1, APC and DVL2 as indicated. Single-channel and merged images are shown. Co-localization with TNKS is indicated with arrows in the merged image of untreated cells. In the merged images, nuclear counterstaining is shown in blue.

**Figure 2 genes-12-01920-f002:**
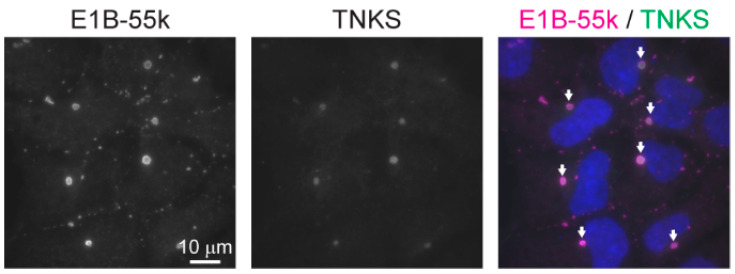
The adenoviral E1B-55k protein present in HEK293 cells co-localize with the large cytoplasmic aggregates. IF analysis (WF) of the localization of the E1B-55k and TNKS proteins in untreated HEK293 cells. In the merged color image, E1B-55k and TNKS co-localization is indicated with arrows and nuclear counterstaining is shown in blue.

**Figure 3 genes-12-01920-f003:**
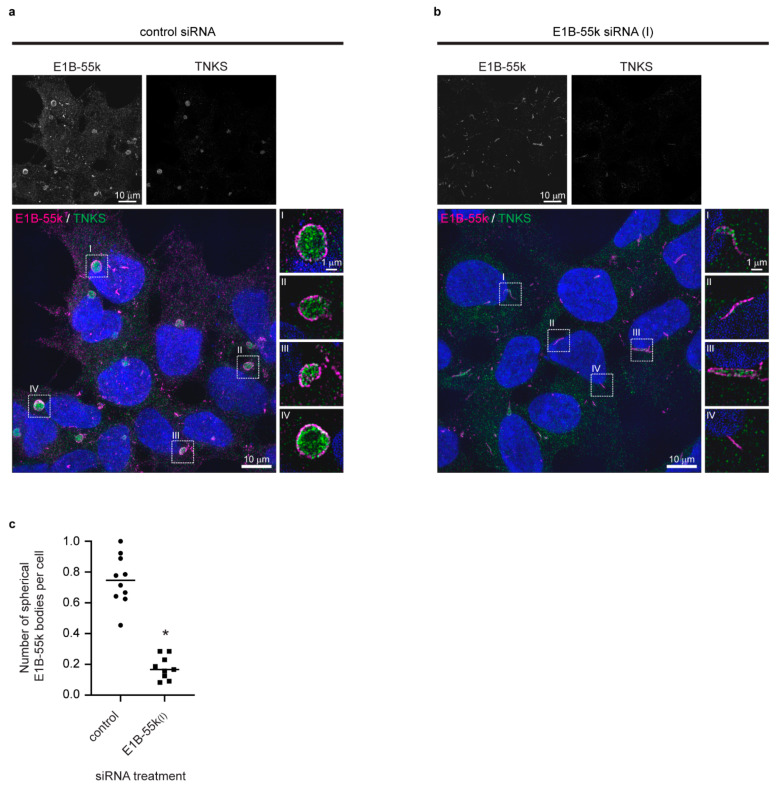
The cytoplasmic aggregates are organized spherical structures which are enclosed by a narrow layer of adenoviral E1B-55k protein. Reduction of E1B-55k protein levels causes the disassembly of the spherical aggregates. Representative SIM images of HEK293 cells treated with (**a**) control or (**b**) E1B-55k siRNA. Upper rows display single-channel images of E1B-55k and TNKS as indicated. In the merged color images, nuclear counterstaining is shown in blue, and enlargements of the stippled boxes are shown to the right (single Z plane). (**c**) Quantification of mean number of cytoplasmic spherical aggregates per cell in control and E1B-55k siRNA-treated cells. Each point represents the mean number of cytoplasmic aggregates per cell in one microscopic view field. For each siRNA treatment, 10 random view fields were quantified. The asterisk indicates a significant difference between control and E1B-55k siRNA-treated cells (*p* < 0.0001, unpaired *t*-test).

**Figure 4 genes-12-01920-f004:**
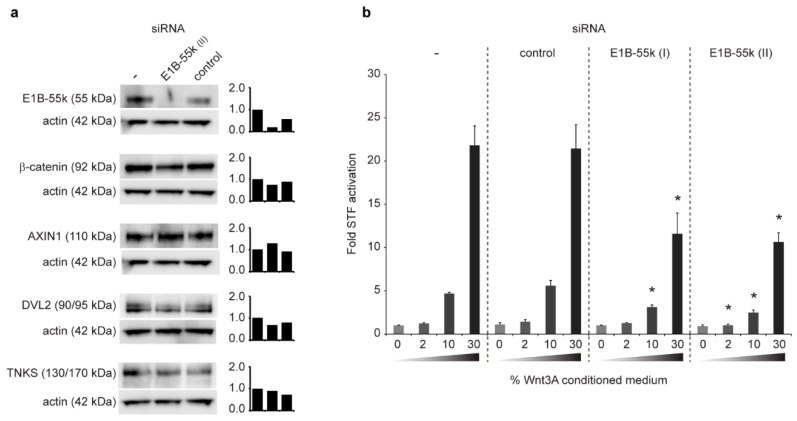
Reduction of E1B-55k protein levels in HEK293 cells causes decreased WNT/β-catenin mediated transcriptional activation upon Wnt3A agonist treatment. (**a**) Western blot analysis of E1B-55k, TNKS, AXIN1 and DVL2 protein levels in untreated (-) and HEK293 cells treated with E1B-55k and control siRNA as indicated. Actin protein levels as loading control are shown below the corresponding bands. The bars shows the quantification of protein levels as determined by calculating the ratio of intensities of the respective protein bands relative to the corresponding actins band. Protein levels relative to untreated cells which where normalized to 1.0 are shown. (**b**) STF reporter gene assay measuring β-catenin mediated transcriptional activation in HEK293-STF/Ren cells treated with none (-), control or two independent E1B-55k siRNA constructs (I and II). The cells were incubated in growth medium supplemented with 0%, 2%, 10% or 30 % Wnt3A CM as indicated. Shown is fold STF activation relative to untreated cells (without siRNA and Wnt3A CM) from three parallel measurements. Error bars represent SD. Asterisk indicate significant difference (one-way ANOVA) relative to corresponding control siRNA-treated cells. For treatment with 2% Wnt3A CM: E1B-55k siRNA (II) *p* = 0.0170; for treatment with 10% Wnt3A CM: E1B-55k siRNA (I) *p* < 0.0001, E1B-55k siRNA (II) *p* < 0.0001; for treatment with 30% Wnt3A CM; E1B-55k siRNA (I) *p* = 0.0003, E1B-55k siRNA (II) *p* = 0.0001.

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
