# Peer review of "The Adenoviral E1B-55k Protein Present in HEK293 Cells Mediates Abnormal Accumulation of Key WNT Signaling Proteins in Large Cytoplasmic Aggregates"

_genes, 2021, doi:10.3390/genes12121920_

Round 1

Reviewer 1 Report

Following comments should be considered to strengthen the claims of the manuscript. After these claims are addressed, I recommend this manuscript for publication.

  1. Line 128 and 130 – LC-MS/MS not LS-MS/MS.
  2. Have the authors performed WB after the administration of Wnt3A CM vs G007-LK to show effects on downstream degradasome protein levels? WB could also be used to see the levels of Axin-2 after the treatment of the CM or the drug. Recommend including WB.
  3. What is the functional relevance of Adenoviral E1B-55K in viral context in general? Is it known to form spherical structures or an artifact of over expression?
  4. Can the authors speculate on the nature of the E1B-55K aggregates.
  5. Wouldn’t one expect some Wnt signaling proteins as hits in the MS?
  6. Suggest quantifying WB 4a.
  7. Can the E1B-55K siRNA phenotype be rescued by over expressing siRNA resistant E1B-55K back to the cells? Strongly recommend doing this experiment to strengthen the conclusions.

Author Response

Thank you for valuable suggestions.

Please see attachment for point-by-point response.

Reviewer 2 Report

The manuscript by Petter Angell Olsen and Stefan Krauss describes new observation, that the adenoviral E1B-55k protein present in HEK293 cells mediates abnormal accumulation of key WNT signaling proteins in large cytoplasmic aggregates and reduction of E1B-55k protein levels leads to disap-385 pearance of the cytoplasmic aggregates, followed by reduced activity of WNT/β-catenin signaling in HEK293 392 cells.

In my opinion, the manuscript is well prepared and provided new knowledge about cell signaling regulation by viral infections.

Some small correction should be made to improve the text:

  1. In Materials and Methods please add napes of suppliers eg. for DMEM, FBS, antibiotics, G007-LK etc.
  2. In line 125 please correct notation of degrees C. 
  3. In line 164 "per cell" is doubled.
  4. In line 285 sentence is incomplete.

Author Response

(The authors gave the same response as above.)
